**Data Availability Statement:** All relevant data are within the paper and its Supporting information files.

# The status of prehospital care delivery for COVID-19 patients in Addis Ababa, Ethiopia: The study emphasizing adverse events occurring in prehospital transport and associated factors

**Ararso Baru**[1,2]*, **Menbeu Sultan**[3], **Lemlem Beza**[4]

1 College of Medicine and Health Sciences, Arbaminch University, Arbaminch, Ethiopia, 2 Slum and Rural Health Initiative-Ethiopia, Addis Ababa, Ethiopia, 3 Department of Emergency Medicine and Critical Care, Saint Paul's Hospital Millennium Medical College, Addis Ababa, Ethiopia, 4 Department of Emergency Medicine and Critical Care, Addis Ababa University, Addis Ababa, Ethiopia

 These authors contributed equally to this work.
* ararsob@gmail.com

## Abstract

### Background

COVID-19 patients may require emergency medical services for emergent treatment and/or transport to a hospital for further treatment. However, it is common for the patients to experience adverse events during transport, even the shortest transport may cause life-threatening conditions. Most of the studies that have been done on prehospital care of COVID-19 patients were conducted in developed countries. Differences in population demographics and economy may limit the generalizability of available studies. So, this study was aimed at investigating the status of prehospital care delivery for COVID-19 patients in Addis Ababa focusing on adverse events that occurred during transport and associated factors.

### Methods

A total of 233 patients consecutively transported to Saint Paul's Hospital Millennium Medical College from November 6 to December 31, 2020, were included in the study. A team of physicians and nurses collected the data using a structured questionnaire. Descriptive statistics were used to summarize data, and ordinal logistic regression was carried out to assess the association between explanatory variables and the outcome variable. Results are presented using frequency, percentage, chi-square, crude and adjusted odds ratios (OR) with 95% confidence intervals.

### Results

The overall level of adverse events in prehospital setting was 44.2%. Having history of at least one chronic medical illness, [AOR3.2 (95%; CI; 1.11–9.53)]; distance traveled to reach destination facility, [AOR 0.11(95%; CI; 0.02–0.54)]; failure to recognize and administer

**Funding:** The authors would like to declare that a small grant was received for this study from the Ethiopian Society of Emergency Medicine Professionals (ESEP), which covered data collection expenses.

**Competing interests:** The authors declare that they have no competing interests.

oxygen to the patient in need of oxygen, [AOR 15.0(95%; CI; 4.0–55.7)]; absent or malfunctioned suctioning device, [AOR 4.0(95%; CI; 1.2–13.0)]; patients handling mishaps, [AOR 12.7(95%; CI; 2.9–56.8)] were the factors associated with adverse events in prehospital transport of COVID-19 patients.

## Conclusions

There were a significant proportion of adverse events in prehospital care among COVID-19 patients. Most of the adverse events were preventable. There is an urgent need to strengthen prehospital emergency care in Ethiopia by equipping the ambulances with essential and properly functioning equipment and trained manpower. Awareness creation and training of transport staff in identifying potential hazards, at-risk patients, adequate documentation, and patient handling during transport could help to prevent or minimize adverse events in prehospital care.

## Background

Similar to other developing countries, there is a growing need for prehospital emergency services in Ethiopia [1]. Although emergency medical services (EMS) in Ethiopia is still in its nascent stages, the EMS in Addis Ababa (the capital city of Ethiopia) is operated by both public and private sectors [2]. The city has public ambulances, dispatch centers, and a toll-free hotlines dedicated to serve COVID-19 patients [3].

COVID-19 patients may require emergency medical services for emergent treatment and/or transport to a hospital for further treatment [4]. Effective transportation requires maintaining the balance between patient and health care providers' safety [5]. However, transporting COVID-19 patients could be risky to both health care professionals and the patient [6–8].

It is common for the patients to experience adverse events during transport, even the shortest transport may cause life-threatening conditions [7, 9–11]. Patient transport can lead to various adverse events such as severe hypotension, decreasing level of consciousness, oxygen desaturation, accidental extubation, poorly or incorrectly placed endotracheal tube, procedural errors, neurological deterioration, medication errors, accidental physical injuries, longer hospital, and intensive care unit (ICU) stay, and death [7, 10, 12–14].

EMS providers' safety can be also compromised during the transport of patients with highly communicable diseases [15]. They are at increased risk of exposure to pathogenic microorganisms, especially when they practice in an environment with limited personal protective equipment (PPE) [9, 15].

Evidence has shown that the fundamental principle of guaranteeing the best available standard emergency care to COVID-19 patients could be challenging during the transport of the patient between facilities [12]. The factors that compromise professionals' and patient's safety during the transport of highly communicable diseases, including COVID-19, are multifaceted. Shortages of personnel and material resources, lack of proper disinfection technique of the ambulance, lack of adequate training on infection prevention, poorly fit PPE, unwillingness to provide care to suspected or confirmed patient with highly infectious disease, procedural failure during patient care, omission of interventions recommended in protocol or commission of interventions outside of the protocol, and accidental injury were among identified reasons for the occurrence of adverse events in prehospital care [7, 9, 16–20].

Most of the studies that have been done on prehospital care of COVID-19 patients were conducted in developed countries [7, 12, 21]. Differences in population demographics, economy, and culture may limit the generalizability of those studies to prehospital transport of COVID-19 patients in Ethiopia. As a result, to the best of our knowledge, there is a dearth of literature on prehospital transport of COVID-19 patients in general and adverse events in prehospital care in particular in Ethiopia. So, this study was aimed at investigating the status of prehospital care delivery for COVID-19 patients in Addis Ababa focusing on adverse events that occurred during transport and their associated factors.

## Methods

### Study location

The study was conducted in Addis Ababa, which is the capital of Ethiopia and the seat for the head office of the African Union. The study was conducted among COVID-19 patients transported by ambulance to the COVID-19 treatment center of St. Paul millennium Medical College (SPMMC) between November 6 to December 31, 2020.

### Study design

The institutional-based cross-sectional study design was carried out to assess the status of prehospital care delivery for COVID-19 patients in Addis Ababa focusing on adverse events that occurred among patients transported to the COVID-19 treatment center of SPMMC.

### Source and study population

The source population was all COVID-19 patients transported by ambulance to treatment centers in Addis Ababa, Ethiopia. The study population was COVID-19 patients transported by ambulance to the COVID-19 treatment center of SPMMC during the data collection period.

### Eligibility criteria

All patients aged 18 years and above, tested positive for COVID-19, and transported by ambulance were included in the study. Meanwhile, patients transported to the treatment center without an ambulance and intra-hospital transports were excluded from the study as they are not part of the study objectives.

### Sample size and sampling procedure

The sample size (n) was determined using single population proportion formula considering the following assumptions: setting the level of confidence (α) at 0.05 (Z (1-α) = 1.96) and the margin of error at 0.05; the proportion of adverse events in a prehospital setting among patients with the life-threatening condition was considered as 16.5% [18]. Considering 10% for missing data and non-response rate, a total of 233 patients were retrospectively followed for the outcome of interest. A total of 233 patients consecutively transported to the COVID-19 treatment center of SPMMC who met eligibility criteria were included in the study.

### Data collection tool and method

Data compilation form was prepared following reviews of the literature to gather information related to the patient, system, and EMS provider. An adverse event was measured by a trigger tool originally designed for helicopter-based emergency care developed by Patterson et al [22], which was also used by Hagiwara and associates for ground-based pre-hospital care [18]. The

tool was modified to meet the study objectives. Data was collected from referral letters, EMS narratives, EMS providers, and patients. Data was collected by groups that constituted general practitioners and nurses (who hold a bachelor of science in Nursing) working in the triage unit of the COVID-19 center.

## Data quality assurance

The qualities of data were assured through the careful design of a comprehensive data collection tool. The validity, practicability, and interpretability of responses for each question on the tool were confirmed by conducting a pilot study on 12 respondents (5% of the sample size) recruited from SPMMC. Based on the feedback from the pilot study, the format and wording of questions were corrected and refined. The respondents recruited for the pilot study were not included into the actual analysis. In addition, training was given to the data collectors and supervisors regarding the study tool. Furthermore, continuous and close supervision of the data collecting procedures, proper categorization, and coding of the data was done. The study investigators and the supervisors checked the completeness and consistency of data daily.

## Data entry, processing, and analysis

The data were checked for completeness and inconsistencies. It was entered, cleaned, and coded using EPI data version 3.1. The entered data were exported to Statistical Package for Social Science (SPSS) version 25 for analysis.

Descriptive statistics such as frequencies, percentages, and cross-tabulations were used to summarize data while the tables were used to present the findings. Pearson chi-square test was used to test the association of each trigger origin with the outcome variable.

Logistic regression models are classified as multinomial, ordinal, and binary logistic regression. When the dependent variable has only two categories, a binary logistic model is appropriate to analyze the association between the dependent variable and a set of explanatory variables. A variable outcome with more than two categories is known as a polytomous outcome [23]. The dependent variable in this study has polytomous outcome categories, which were collected on ordinal responses viz; no adverse events, adverse events with potential for harm, and adverse events with harm identified. It has been reported that converting ordinal outcomes into binary categories results in a loss of information [24]. Therefore, ordinal regression was used to examine the association between dependent and independent variables. Ordinal logistic regression refers to the case where the dependent variable has an order. The generalized ordered logit model is one of the most commonly used ordinal logistic regression models and it is an appropriate model for the analysis of an ordinal outcome from survey data [25]. Each variable was first tested individually to see if the requirements of the proportional odds (PO) assumption of ordinal logistic regression were satisfied. As PO assumptions were satisfied for each explanatory variable, the ordinal proportional odds model was applied in this study. The level of statistical significance was set at $p < 0.05$. The odds ratio with 95% confidence intervals, which was calculated from bivariate and multivariable ordinal logistic regression analysis were reported in the study.

## Ethical consideration

Ethical clearance was obtained from SPMMC ethical review board (Certificate of ethical clearance No: P.M. 23/7/2). The purpose of the study was explained to the director of the treatment center and concerning bodies. Confidentiality of the information obtained from each respondent was maintained. Considering highly contagious nature of the virus, the risks of producing biohazard as the materials on which consent is recorded could be contaminated, and low

literacy population, verbal consent was secured from each respondent before proceeding to data collection and it was approved by IRB. Verbal consent was suggested as acceptable alternative to written consent to reduce the spread of SARS CoV-2 by previous literatures [26, 27].

### Operational definitions

**Adverse events.** Are defined as an event in the EMS that is a harmful or potentially harmful event occurring during the continuum of EMS care that is potentially preventable and thus independent of the progression of the patient's condition [28].

**No adverse events.** A case where a trigger was selected (e.g., cardiac arrest during transport), but no AE identified after full review [22].

**Harmful adverse events.** An action or omission that led to injury or harm regardless of severity [22].

**Potentially harmful adverse events.** An action that may lead to injury or harm but there is no evidence that an injury or harm occurred [22].

## Results

### Baseline characteristics of the study participants

A total of 233 COVID-19 patients transported by ambulance to SPMMC were included in the study. Table 1 presents basic information that summarizes the demographic characteristics of COVID-19 patients and EMS providers. From the total 233 COVID-19 patients included in the study, nearly two-thirds (63.5%) of them were male. Of all patients included in this study, 67(28.8%) were aged 51–65 years followed by 18–35 years 61(26.2%). Nearly eight in ten 182 (78.1%) EMS providers that transported COVID-19 patients to SPMMC hold a bachelor of science in nursing. About half 119(51.1%) of the EMS providers had 3–6 months of work experience in EMS. The majority of the patients arrived at the COVID-19 treatment center from a distance of ≤km from the center and only 9% of the total patients arrived at the COVID-19 treatment center within ≤8 minutes of their referral. The mean prehospital time was 27.2 minutes with a standard deviation of 18.9.

### Clinical characteristics of the study participants

As shown in Table 2, slightly over one-third 83(35.6%) of the patients had mild COVID-19 before transport followed by moderate and severe COVID-19, 79(33.9%), and 42(18.0) respectively. Meanwhile, on arrival at to treatment center 69(29.6%) had mild, 68(29.2%) had moderate, and 51(21.9%) had severe COVID-19.

About 46% of the patients had at least one comorbid medical illness. Diabetic Mellitus 56 (24.0%) was the commonest comorbid medical illness followed by hypertension and renal diseases, which accounts for 41(17.6%) and 37(15.9%) respectively.

### Level of adverse events in prehospital transport

Table 3 shows that 55.8% of the patient had no adverse events, 34.2% had potentially harmful adverse events while 9.4% experienced harmful adverse events. Moreover, the levels of adverse events in prehospital transport of COVID-19 patients was 103(44.2%).

### Patients, EMS providers, and system-related encountered events during prehospital transport

The results in Table 4 depicted that thirty-five (15%) of the total COVID-19 patients transported to the treatment center initially refused transport while 29(12.4%) refused treatment

**Table 1. Baseline characteristics of COVID-19 patients transported by ambulance to SPMMC, November to December 2020, Addis Ababa, Ethiopia.**

| Variables | Frequency (n = 233) | Percentage |
|---|---|---|
| **Sex** | | |
| Male | 148 | 63.5 |
| Female | 85 | 36.5 |
| **Age** | | |
| 18–35 | 61 | 26.2 |
| 36–50 | 57 | 24.5 |
| 51–65 | 67 | 28.8 |
| >65 | 48 | 20.6 |
| **Qualification of EMS provider** | | |
| EMT | 17 | 7.3 |
| Diploma Nurse | 29 | 12.4 |
| BSc Nurse | 182 | 78.1 |
| Others | 5 | 2.1 |
| **EMS experiences in months** | | |
| <3 | 41 | 17.6 |
| 3–6 | 119 | 51.1 |
| 7–12 | 37 | 15.9 |
| >12 | 36 | 15.5 |
| **Prehospital time (in minutes) (Mean = 27.2; SD = 18.9** | | |
| ≤8 | 21 | 9.0 |
| 9–15 | 69 | 29.6 |
| 16–30 | 69 | 29.6 |
| 31–45 | 37 | 15.9 |
| 46–60 | 22 | 9.4 |
| ≥60 | 15 | 6.4 |
| **Distance traveled to reach receiving facility** | | |
| ≤5.0 km | 132 | 56.7 |
| 5.1–10.0 km | 49 | 21.0 |
| 10.1–15.0 km | 34 | 14.6 |
| >15.0km | 18 | 7.7 |

specified on protocol, 2(9.4%) patient either discontinued or tried to discontinue an ongoing therapy and 36 (15.5%) patient took action that resulted or may result in harm to themselves or others.

Over one-fourth 66(28.3%) of the patients required oxygen but did not get oxygen administration. In addition, oxygen supply was diminished during transport in 27(11.6%) of the total transported patients (Table 4).

There were shortages of some resources that were important to patient's care but not available or malfunctioned during transport. Table 4 shows that 25(10.7%) needed suctioning devices, 61(26.2%) were needed a pulse oximeter, however, they were not present or malfunctioned during COVID-19 patients' transport. In addition, a lack of PPE was reported in 38 (16.3%) of the total cases.

## Trigger origin of the case containing adverse events

It was shown in Table 5 that documentation trigger was the most common identified trigger origin for adverse events in prehospital care among COVID-19 patients. About one in ten

**Table 2. Distribution of COVID-19 severity and comorbid medical illness among COVID-19 patients transported to SPMMC by ambulance, November to December 2020, Addis Ababa, Ethiopia.**

| Variables | Categories | Frequency (n = 233) | Percentage |
|---|---|---|---|
| COVID-19 severity before transport | Mild | 83 | 35.6 |
| | Moderate | 79 | 33.9 |
| | Severe | 42 | 18.0 |
| | Critical | 21 | 9.0 |
| | Unknown | 8 | 3.4 |
| COVID-19 severity after transport | Mild | 69 | 29.6 |
| | Moderate | 68 | 29.2 |
| | Severe | 51 | 21.9 |
| | Critical | 45 | 19.3 |
| Presence of at least one comorbid illness | Yes | 107 | 45.9 |
| | No | 126 | 54.1 |
| Diabetic Mellitus | Yes | 56 | 24.0 |
| | No | 177 | 76.0 |
| Hypertensive | Yes | 41 | 17.6 |
| | No | 192 | 82.4 |
| Heart disease | Yes | 18 | 7.17 |
| | No | 215 | 92.3 |
| Renal disease | Yes | 37 | 15.9 |
| | No | 196 | 84.1 |
| Asthmatic | Yes | 13 | 5.6 |
| | No | 220 | 94.4 |
| Immunocompromised | Yes | 9 | 3.9 |
| | No | 224 | 96.1 |
| Other comorbid illness | Yes | 20 | 8.6 |
| | No | 213 | 91.4 |

COVID-19 patients transported by ambulance had either missed, incomplete or unclear documentation of prehospital care provided to them. The worsening trend in the patient's hemodynamic status was the second most common trigger origin of the case containing adverse events followed by failure to perform intervention within the protocol or performing intervention outside of the protocol.

## Factors associated with adverse events in prehospital COVID-19 patients transport

The findings from the multivariate analysis showed a statistically significant association between the total distance traveled to reach receiving facility and adverse events in prehospital care, [AOR 0.11(95%; CI; 0.02–0.54)]. Implying that COVID-19 patients who traveled 10 to 15

**Table 3. Distribution of adverse events among COVID-19 patients transported by ambulance to SPMMC, November to December 2020, Addis Ababa, Ethiopia.**

| Variables | Frequency (n = 233) | Percentage |
|---|---|---|
| No adverse events | 130 | 55.8 |
| Adverse events with potential for harm | 81 | 34.8 |
| Adverse events with harm identified | 22 | 9.4 |
| **Total** | 233 | 100.0 |

**Table 4. Distribution of patients, EMS providers, and system-related encountered events in prehospital transport of COVID-19 to SPMMC, November to December 2020, Addis Ababa, Ethiopia.**

| Variables | Categories | Frequency (n = 233) | Percentage |
|---|---|---|---|
| The patient refused transport to the treatment centre | Yes | 35 | 15.0 |
| | No | 198 | 85.0 |
| The patient refused treatment specified in the protocol | Yes | 29 | 12.4 |
| | No | 204 | 87.6 |
| The patient discontinued or tried to discontinue an ongoing therapy | Yes | 22 | 9.4 |
| | No | 211 | 90.6 |
| The patient took action that results or may result in harm to themselves or others | Yes | 36 | 15.5 |
| | No | 197 | 84.5 |
| Failure to recognize and administer oxygen to the hypoxic patient | Yes | 66 | 28.3 |
| | No | 167 | 71.7 |
| Delay in patient care due to delay in receiving or referring facility | Yes | 70 | 30.0 |
| | No | 163 | 70.0 |
| Oxygen supply diminished during transport | Yes | 27 | 11.6 |
| | No | 206 | 88.4 |
| The suctioning device was requested but not present or malfunctioned | Yes | 25 | 10.7 |
| | No | 208 | 89.3 |
| Pulse oximeter was needed but not present | Yes | 61 | 26.2 |
| | No | 172 | 73.8 |
| Failure to provide care due to lack of PPE | Yes | 38 | 16.3 |
| | No | 194 | 83.7 |
| Patients handling mishaps | Yes | 13 | 5.6 |
| | No | 220 | 94.4 |
| Delay in patient transport due to scarcity of ambulance | Yes | 20 | 8.6 |
| | No | 213 | 91.4 |

kilometers to reach receiving facility were 89% less likely to experience adverse events in prehospital care compared to those who traveled beyond 15 kilometers by ambulances. (Table 6).

As shown in Table 6, the previous history of at least one chronic medical illness independently predicts adverse events in prehospital care, [AOR3.2 (95%; CI; 1.11–9.53). COVID-19 patients who had at least one comorbid medical illness were 3.2 times at increased risks of experiencing adverse events in prehospital care than those without a history of comorbid medical illness.

The study also found that failure to recognize and administer oxygen to the patient in need of oxygen (hypoxic patients) were associated with adverse events in prehospital COVID-19 patients transport in both bivariate and multivariate analyses, [COR12.3(95%; CI;6.40–23.78)] and [AOR 15.0(95%; CI; 4.0–55.7)] respectively. It means that COVID-19 patients who required oxygen but left unrecognized during prehospital transport were fifteen times more likely at increased risks of experiencing adverse events compared to other transported patients.

Absent or malfunctioned suctioning device was independently associated with adverse events in prehospital care of COVID-19 patients, [AOR 4.0(95%; CI; 1.2–13.0)]. Implying that COVID-19 patients who required suctioning but the device was absent or malfunctioned were four times more likely at increased risks of adverse events in prehospital care than their counterparts. (Table 6).

The present study also identified that patients handling mishaps which includes dropping of the patient because of the malfunctioned stretcher, or during manual lifting, or while

**Table 5. Distribution of trigger origin of the case containing adverse events.**

| Triggers | Category | Triggers n (%) | Adverse events | | | Pr Chi-square | p-value |
|---|---|---|---|---|---|---|---|
| | | | No n (%) | Potential for harm n(%) | harm identified n(%) | | |
| Missing, incomplete, or unclear documentation | Yes | 208(89.3) | 120 (51.5) | 68(29.2) | 20(8.6) | 3.7 | 0.157 |
| | No | 25(10.7) | 10(4.3) | 13(5.6%) | 2(0.9) | | |
| Response time exceeds accepted standards | Yes | 66(28.3) | 32 (13.7) | 22(9.4) | 12(5.2) | 8.4 | 0.015* |
| | No | 167(71.7) | 98 (42.1) | 59(25.3) | 10(4.3) | | |
| Injury to the patient or team member during patient encounter/ transport | Yes | 35(15.0) | 9(3.9) | 16(6.9) | 10(4.3) | 24.1 | <0.001* |
| | No | 198(85.0) | 121 (51.9) | 65(27.9) | 12(5.2) | | |
| Requested additional resources | Yes | 38(16.3) | 13(5.6) | 17(7.3) | 8(3.4) | 11.6 | 0.003* |
| | No | 195(83.7) | 117 (50.2) | 64(27.5) | 14(6.0) | | |
| A worsening trend in patient hemodynamic or mental status | Yes | 82(35.2) | 28 (12.0) | 36(15.5) | 18(7.7) | 34.6 | <0.001* |
| | No | 151(64.8) | 102 (43.8) | 45(19.3) | 4(1.7) | | |
| Cardiac arrest during transport | Yes | 2(0.9) | 0(0.0) | 2(0.9) | 0(0.0) | 3.8 | 0.151 |
| | No | 231(99.1) | 130 (55.8) | 79(33.9) | 22(9.4) | | |
| Use of cardioversion or defibrillation or advanced airway attempt or surgical airway, intraosseous access or chest decompression or chest tube | Yes | 0(0.0) | 0(0.0) | 0(0.0) | 0(0.0) | - | - |
| | No | 233(100.0) | 130 (55.8) | 81(34.8) | 22(9.4) | | |
| Failure of any intervention or procedure during patient care | Yes | 21(9.0) | 2(0.9) | 15(6.4) | 4(1.7) | 20.0 | <0.001* |
| | No | 212(91.0) | 128 (54.9) | 66(28.3) | 18(7.7) | | |
| Use of blood products or vasopressors or inotropes | Yes | 8(3.4) | 3(1.3) | 2(0.9) | 3(1.3) | 7.6 | 0.022* |
| | No | 225(96.6) | 127 (54.5) | 79(33.9) | 19(8.2) | | |
| Commission of an intervention that appears to be outside of protocol, or omission of an intervention that is within the standard of care | Yes | 87(37.3) | 38 (16.3) | 36(11.2) | 13(15.5) | 9.8 | 0.007* |
| | No | 146(62.7) | 92 (39.5) | 45(19.3) | 9(3.9) | | |
| Medication error | Yes | 28(12.0) | 11(4.7) | 13(5.6) | 4(1.7) | 3.6 | 0.166 |
| | No | 205(88.0) | 119 (51.1) | 68(29.2) | 18(7.7) | | |

*Indicate statistically significant association.

transferring to or from or the stretcher were associated with adverse events in prehospital transport of COVID-19 patients on multivariate analysis, [AOR 12.7(95%; CI; 2.9–56.8)] (Table 6).

Moreover, the bivariate analysis found that COVID-19 patients with diabetic Mellitus, [COR 4.3(95%; CI; 2.35–7.76)]; hypertensive COVID-19 patients, [COR 4.3(95%; CI; 2.35–7.76)]; patient refused treatment specified in the protocol, [COR 3.5(95%; CI; 1.73–7.01)], and the patient who discontinued or tried to discontinue an ongoing treatment were associated with adverse events in prehospital transport. However, the aforementioned variables were no more statistically significant after adjusting for potential confounding variables. Yet, they have public health and clinical relevance as shown in Table 6 on multivariate analysis.

**Table 6. Factors associated with prehospital adverse events among COVID-19 patients transported to SPMMC, November to December 2020, Addis Ababa, Ethiopia.**

| Variables | Adverse events | | | COR (95%CI) | AOR (95%CI) |
|---|---|---|---|---|---|
| | No | Potential for harms | Harm identified | | |
| **Distance traveled to reach SPMMC in km** | | | | | |
| ≤5.0 | 76 | 48 | 8 | 0.25(0.09–0.67)** | 0.40(0.10–1.52) |
| 5.1–10.0 | 26 | 16 | 7 | 0.34(0.12–0.99)* | 0.51(0.12–2.21) |
| 10.1–15.0 | 22 | 11 | 1 | 0.18(0.06–0.58)** | 0.11(0.02–0.54)* |
| >15.0 | 6 | 6 | 6 | Ref. | Ref. |
| **At least one chronic illness** | | | | | |
| Yes | 38 | 53 | 16 | 2.7(1.80–3.95)*** | 3.2(1.11–9.53)* |
| No | 92 | 28 | 6 | Ref. | Ref. |
| **Diabatic Mellitus** | | | | | |
| Yes | 16 | 29 | 11 | 4.3(2.35–7.76)*** | 1.9(0.64–5.6) |
| No | 114 | 52 | 11 | Ref. | Ref. |
| **Hypertensive** | | | | | |
| Yes | 10 | 22 | 9 | 4.6(2.40–9.04)*** | 1.4(0.49–3.91) |
| No | 120 | 59 | 13 | Ref. | Ref. |
| **The patient refused transport to the treatment center** | | | | | |
| Yes | 10 | 23 | 2 | 1.5(1.11–1.96)** | 1.5(0.26–9.1) |
| No | 120 | 58 | 20 | Ref | Ref. |
| **The patient refused treatment specified in the protocol** | | | | | |
| Yes | 5 | 22 | 2 | 3.5(1.73–7.01)*** | 3.9(0.61–25.9) |
| No | 125 | 59 | 20 | Ref. | Ref. |
| **The patient discontinued or tried to discontinue an ongoing treatment** | | | | | |
| Yes | 4 | 17 | 1 | 3.0(1.40–6.55)** | 2.3(0.41–12.9) |
| No | 126 | 64 | 21 | Ref. | Ref. |
| **Failure to recognize and administer oxygen to the hypoxic patient** | | | | | |
| Yes | 10 | 40 | 16 | 12.3(6.40–23.78)*** | 15.0(4.0–55.7)*** |
| No | 120 | 41 | 6 | | Ref. |
| **Oxygen supply diminished during transport** | | | | | |
| Yes | 7 | 14 | 6 | 3.9(1.80–8.36)** | 2.1(0.66–6.43) |
| No | 123 | 67 | 16 | Ref. | Ref. |
| **The suctioning device was needed but not present or malfunctioned** | | | | | |
| Yes | 9 | 7 | 9 | 4.1(1.76–9.67)** | 4.0(1.2–13.0)* |
| No | 121 | 74 | 13 | Ref. | Ref. |
| **Patients handling mishaps** | | | | | |
| Yes | 5 | 3 | 5 | 3.8(1.19–11.95)* | 12.7(2.9–56.8)** |
| No | 125 | 78 | 17 | | Ref. |

*p<0.05,

**p<0.01,

***p<0.001.

## Discussions

The levels of adverse events in this study were 44.2%, with 34.8% classified as the potential for harm and 9.4% as harm identified (Table 3). This finding was higher than that of the study conducted in Sweden using the same tool with this study. The Swedish study reported a prehospital adverse events rate of 4.3% among all transported patients and 16.5% among patients

with life-threatening conditions [18]. The difference could be attributable to several factors. Prehospital care in Ethiopia is less developed and operating in an environment with a shortage of resources including trained manpower, essential equipment, and emergency drugs [29–31]. In addition, the present study focused only on prehospital care of COVID-19 patients while the Swedish study focused on all patients transported by ambulance regardless of their diseases although there was no COVID-19 at the time of the study. Evidence has shown that healthcare workers are less willing to provide care for patients with infectious diseases compared to other disasters or mass casualty incidents because of the associated risk of exposure to communicable diseases [17]. So, there could be possibilities of negligence in the patient care in the present study due to fear of exposure to COVID-19 compared to the Swedish study.

It has been recommended that an ideal ambulance response time should be equivalent to less than 8 minutes as it is associated with adverse patients outcomes [32]. Ambulance response time in this study was greater than 8 minutes in 28.3% of the transported patients. Although the present findings were still not adhered to the recommended standard, our finding was far better than the results of the previous study conducted in Addis Ababa before the COVID-19 outbreak [30]. Several possibilities may have contributed to the disparities. Unlike the situation before COVID-19, as a response to the outbreak emergency task force was established to provide rapid communication and handling of COVID-19 cases in Ethiopia [33]. In addition, the country has activated toll-free hotlines and ambulance services dedicated to COVID-19 cases [3, 34].

The study found that missing, incomplete, or unclear prehospital documentation was the most common trigger origin for a case containing adverse events. It was occurred in about nine in ten COVID-19 patients transported by ambulance. Of which, nearly 30% had cases containing adverse events with potential for harms and about 9% had adverse events with harm identified (Table 5). Similarly, previous studies reported a high prevalence of missed or incomplete prehospital documentation with adverse outcomes [18, 35].

Errors of omission of an intervention that is within the national standard of care or commission of an intervention that appears to be outside of protocol were identified in 37.3% of prehospital transported COVID-19 cases in this study (Table 5). Evidence has shown that both errors of omission and commission are associated with events and adverse outcomes [19, 20]. Several possibilities may have contributed to the occurrence of errors of commission or omission in the present study. Fear of contracting COVID-19, absence of essential resources to provide emergency medical services, and shortage of personal protective equipment may lead to the omission of an intervention that is within the standard of care. In addition, lack of a clearly stated EMS model, viz; Franco-German (which is 'delay and treat') or Anglo-American model (which is 'scoop and run' response), to guide the action of EMS providers in Ethiopia may have contributed to omission or commission of procedural errors in prehospital care.

It has been suggested that contingency plans including having additional PPE available for all transport personnel should be in place before transport COVID-19 patients [36]. However, this study found that failure to provide prehospital care due to lack of PPE was reported in about 16% of transported COVID-19 patients (Table 4). Although a statistically significant association was not observed in the present study, it has been suggested that maintaining adequate types and volumes of personal protective equipment are crucial factors that affect both safeties of the patients and transport personnel [37].

The proportion of transport refusal in the present study was 15.0% (Table 4). The levels of prehospital transport refusal in this study were slightly lower (19.9%) than the results of transport refusal incidents reported from the study conducted in Israel amid the COVID-19 pandemic [38]. The difference could be explained by the fact that the study from Israel included

all medical emergency incidents regardless of the causes while the present study focused on transports of COVID-19 patients alone. The fear of contracting COVID-19 infection at a hospital in non-infected patients seems to have caused transport refusal in the study conducted in Israel compared to this study. On the contrary, the fear of spreading COVID-19 to an uninfected loved one by staying at home may have caused less transport refusal in the present study compared to a study conducted in Israel as the latter study involved the transport of all medical emergency during the COVID-19 outbreak.

This study demonstrates that previous history of at least one chronic medical illness was independently associated with adverse events in prehospital care (Table 5). The existence of multiple comorbid illnesses has long been identified as key components in predicting adverse events in prehospital care [39].

The study has shown that adverse events may occur while transferring the patient to or from stretcher or while ambulance stretcher operation, and may result in significant injury to the patient [40]. In this study, we found that patients handling mishaps were associated with adverse events in prehospital transport of COVID-19 patients after adjusting for potential confounders (Table 6). Therefore, it is of utmost importance to consider the maximum precaution while handling patients especially during manual lifting of the patients, or while transferring to or from or the stretcher to prevent or minimize patients handling mishaps.

The previous study has reported a statistically significant association between prehospital blood oxygen saturation level and in-hospital adverse outcomes of COVID-19 patients [41]. Although we did not assess the association of in-hospital outcomes with prehospital oxygen saturation status in the present study, our finding showed that failure to recognize and administer oxygen to hypoxic patients was independently associated with adverse events in prehospital care among COVID-19 patients. Oxygen administration is one of the recommended interventions in the prehospital setting for COVID-19 patients in Ethiopia. Independent of COVID-19, the omission of an intervention that is within the standard of care was reported as a significant predictor of adverse events in the prehospital setting [18].

The results from unadjusted analysis in this study showed a statistically significant association between transport refusal and prehospital adverse events. However, our finding was no more statistically significant after adjusting for potential confounders (Table 6). Transport refusal was previously reported as a predictor of clinical deterioration and adverse outcomes in prehospital care [38]. Further studies are needed to confirm the association between transfer refusal and adverse events in prehospital care in Ethiopia.

## Limitations

The study has several limitations. First, it was a cross-sectional study, with the usual inherent limitations. Second, some variables such as procedural errors and medication errors were assessed through self-report or peer report so that there is a possibility of underreporting in the present study. Previous studies showed that underreporting of procedural errors and adverse events by EMS providers were frequent [20]. Third, the present study is conducted in Addis Ababa which has relatively better EMS compared to other places in Ethiopia. Thus, our study has geographical limitations to generalize the findings. Lastly, children were not included in the study, therefore, our results might not apply to a pediatric population.

Despite the aforementioned limitations, to the best of our knowledge, this study was the first of its kind to describe the prehospital status of COVID-19 patients focusing on adverse events in prehospital care in Ethiopia.

### Generalizability

Although this study has some limitations, none of them affected our main findings. However, further studies with study designs other than cross-sectional (i.e. Case-control study with propensity score matching, consideration of bootstrap simulation for logistic regression), different populations, and geographic areas are warranted to verify the generalizability our findings to other parts of Ethiopia and low income countries. In nutshell, our findings could be generalizable to other low income countries as long as considerations given to the context of the study setting, methods and limitations described in this study.

## Conclusions

There were a significant proportion of adverse events in prehospital care among COVID-19 patients. Most are the consequences of problems related to EMS providers and the health care system, and they were preventable. Distance of transport destination from the patient, history of at least one comorbid medical illness, failure to recognize and administer oxygen to a hypoxic patient, absent or dysfunctional suctioning device, and patients handling mishaps were the factors independently associated with adverse events in COVID-19 patients transport in this study. There is an urgent need to strengthen prehospital emergency care in Addis Ababa by equipping the ambulances with essential and properly functioning equipment and trained manpower. Awareness creation and training of transport staff in identifying potential hazards, at-risk patients, adequate documentation, and patient handling during transport could help to prevent adverse events in prehospital care.

## Supporting information

**S1 File.**
(DOCX)

**S1 Data.**
(SAV)

## Acknowledgments

Our deepest gratitude goes to the administration of Saint Paul's Hospital COVID-19 Treatment Center for allowing us to conduct this study in their facility.

## Author Contributions

**Conceptualization:** Ararso Baru, Menbeu Sultan, Lemlem Beza.

**Data curation:** Ararso Baru.

**Formal analysis:** Ararso Baru.

**Funding acquisition:** Ararso Baru, Menbeu Sultan, Lemlem Beza.

**Investigation:** Ararso Baru, Menbeu Sultan, Lemlem Beza.

**Methodology:** Ararso Baru, Menbeu Sultan, Lemlem Beza.

**Project administration:** Ararso Baru.

**Resources:** Ararso Baru, Menbeu Sultan, Lemlem Beza.

**Software:** Ararso Baru.

**Supervision:** Ararso Baru, Menbeu Sultan, Lemlem Beza.

**Validation:** Ararso Baru, Menbeu Sultan, Lemlem Beza.

**Visualization:** Ararso Baru, Menbeu Sultan, Lemlem Beza.

**Writing – original draft:** Ararso Baru.

**Writing – review & editing:** Ararso Baru, Menbeu Sultan, Lemlem Beza.

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
