## [Decision Letter · Decision Letter 0]

9 Aug 2021

PONE-D-21-17807

The status of prehospital care delivery for COVID-19 patients in Addis Ababa, Ethiopia: The study emphasizing adverse events occurring in prehospital transport and associated factors

PLOS ONE

Dear Dr. Ararso Baru,

Thank you for submitting your manuscript to PLOS ONE. After careful consideration, we feel that it has merit but does not fully meet PLOS ONE’s publication criteria as it currently stands. Therefore, we invite you to submit a revised version of the manuscript that addresses the points raised during the review process.

ACADEMIC EDITOR:

Interesting study that explores a field where adequate information is lacking mainly in a Developing Countries.  It underlinines how the cornerstones of the reduction of adverse events are: awareness creation and training of transport staff, adequate documentation, and patient handling during transport.

However, to be publishable on PLOSONE it is necessary that the Authors answer the following points:

To respond accurately to the Reviewers’ requestsTo improve the methodology of the proposed studyTo highlight how the results of the study can be valid in contexts other than the one where the study was carried out.Please provide the number of the Ethical clearance obtained from St. Paul Millennium Medical College (SPMMC) ethical review board and  explain the reasons why verbal consent was authorized by the ethics committee and written consent was not required.In the abstract (Results) correct: “were independently were associated with adverse events in prehospital transport of COVID-19 patients”   were is repeated 2 times.

The decision is justified on PLOS ONE’s publication criteria.  

We look forward to receiving your revised manuscript.

Kind regards,

Filomena Pietrantonio

Academic Editor

PLOS ONE

Journal Requirements:

2. Please include additional information regarding the survey or questionnaire used in the study and ensure that you have provided sufficient details that others could replicate the analyses. For instance, if you developed a questionnaire as part of this study and it is not under a copyright more restrictive than CC-BY, please include a copy, in both the original language and English, as Supporting Information. If the original language is written in non-Latin characters, for example Amharic, Chinese, or Korean, please use a file format that ensures these characters are visible.

3. Please state whether you validated the questionnaire prior to testing on study participants. Please provide details regarding the validation group within the methods section.

4. For more information on PLOS ONE's expectations for statistical reporting, please see https://journals.plos.org/plosone/s/submission-guidelines.#loc-statistical-reporting. Please update your Methods and Results sections accordingly.

5. Please amend your current ethics statement to address the following concerns: Please explain why written consent was not obtained, how you recorded/documented participant consent, and if the ethics committees/IRBs approved this consent procedure.

Additional Editor Comments:

Interesting study that explores a field where adequate information is lacking mainly in a Developing Countries. It underlinines how the cornerstones of the reduction of adverse events are: awareness creation and training of transport staff, adequate documentation, and patient handling during transport.

However, to be publishable on PLOSONE it is necessary that the Authors answer the following points:

1. To respond accurately to the Reviewers’ requests

2. To improve the methodology of the proposed study

3. To highlight how the results of the study can be valid in contexts other than the one where the study was carried out.

4. Please provide the number of the Ethical clearance obtained from St. Paul Millennium Medical College (SPMMC) ethical review board and explain the reasons why verbal consent was authorized by the ethics committee and written consent was not required.

5. In the abstract (Results) correct: “were independently were associated with adverse events in prehospital transport of COVID-19 patients” were is repeated 2 times.

Reviewers' comments:

Reviewer's Responses to Questions

**Comments to the Author**

1. Is the manuscript technically sound, and do the data support the conclusions?

Reviewer #1: Yes

Reviewer #2: Yes

2. Has the statistical analysis been performed appropriately and rigorously? 

Reviewer #1: Yes

Reviewer #2: Yes

3. Have the authors made all data underlying the findings in their manuscript fully available?

Reviewer #1: Yes

Reviewer #2: No

4. Is the manuscript presented in an intelligible fashion and written in standard English?

Reviewer #1: Yes

Reviewer #2: Yes

5. Review Comments to the Author

Reviewer #1: The authors did a good job in describing factors related to adverse effects occurring during pre-hospital transportation of COVID patients in Addis Abeba. They adopted a retrospective deisgn whose limitations were recognised. There are doubts on the generalizability of the study, as the authors themselves acknowledged. The data are provided, and their analysis methodology is sound. I believe the paper can be published after some minor corrections. Here are my suggestions for the authors.

1) - Introduction - EMT is not always performed the same way in all countries. Some countries for instance do not allow private companies to perform EMT, others do, and not all countries have universal and free EMT coverage. Since it cannot be expected by a reader to know such details for every country, and even in the same country differences can arise due to local regulations, please expand this section with a brief description of involved EMT system charactesitics.

2) - Results - Tables 1, 2 and 4 report a descriptive statistics for the sample population, along with a Pearson chi-square value and a p-value. According to the Methods section, "Pearson chi-square test was used in the bivariate analysis to test the association of each independent variable with the outcome variable"; P-value is the probability of observing a value as extreme as the test statistic if the null hypothesis were true (Cfr https://www.ncbi.nlm.nih.gov/pmc/articles/PMC4111019/). However, no association was shown in either tables 1 and 2 nor in text, and no alternative hypotesis was generated either in tables not in text, concerning population characteristics. Please clarify the meaning of such two values, if any, or remove them for Tables 1, 2 and 4. If some statistica

3) - Conclusions - Directly related to point 1, if the involved EMT system is not universal and free (toll-free hotlines are mentioned but it is not clear if EMT are also toll-free), please discuss if this may allow selection bias in population sampling (including considerations on populations literacy), highlighting how the study design and results can (or cannot) be generalized for countries with similar population, economy, and EMT systems.

Reviewer #2: this is an interesting piace of research that deserves publication also for the originality and relevance of the research question.

I have some suggestions:

- in the logistic regression I would suggest to include a bootstrap simulation in order to test the robustness of results

- a further development of this paper should be to adopt a quasi experimental approach where case and controls are identified (i.e. different approaches of taking in charge patients…) and performing a propensity score matching artificial sampling. You should include this further development eventually in the discussion?

6. PLOS authors have the option to publish the peer review history of their article (what does this mean?). If published, this will include your full peer review and any attached files.

Reviewer #1: **Yes: **Antonio Vinci

Reviewer #2: No

---

## [Author Response · Author response to Decision Letter 0]

16 Aug 2021

Response to editor

1. The revised manuscript is corrected according to PLOS ONE requirements

2. The survey questionnaire is attached as supporting information

3. We were requested by editor to provide detail information on validation of the tool and it is included under data quality assurance part of the methods section in the revised manuscript

4. We were requested to provide IRB approval number and amend ethical statements and we did it accordingly

Response to reviewer 1

1. We were requested to provide further explanation on the type of EMS involved under introduction part and we did it accordingly.

2. Based on comments given by reviewer one, we modified table 1, 2 and 4, and marked the change made into the tables. 

3. We are requested to include generalizability of our findings and we included into the revised manuscript. 

Response to reviewer 2

1. The reviewer suggested on the further development of the study. Based on the suggestion, we recognized limitations of the study and suggested bootstrap simulation, and case-control study with propensity score matching for the further development of the study.

---

## [Decision Letter · Decision Letter 1]

20 Dec 2021

PONE-D-21-17807R1The status of prehospital care delivery for COVID-19 patients in Addis Ababa, Ethiopia: The study emphasizing adverse events occurring in prehospital transport and associated factorsPLOS ONE

Dear Dr. Ararso Baru,

Thank you for submitting your manuscript to PLOS ONE. After careful consideration, we feel that it has merit but does not fully meet PLOS ONE’s publication criteria as it currently stands. Therefore, we invite you to submit a revised version of the manuscript that addresses the points raised during the review process.

ACADEMIC EDITOR:The Authors addressed all comments and improved their manuscript in large part. 

However, in order for the paper to be published, the Authors are asked to respond appropriately to the comments of Reviewer 1, in particular,the "Generalizability" section.  Please expand this section, giving a more detailed explanation (i.e. providing examples or evidence of how study findings can - or can not - be applyed to a different setting).  The decision is justified on PLOS ONE’s publication criteria. 

We look forward to receiving your revised manuscript.

Kind regards,

Filomena Pietrantonio

Academic Editor

PLOS ONE

Journal Requirements:

Additional Editor Comments:

The authors addressed all comments and improved their manuscript in large part.

However, in order for the paper to be published, the Authors are asked to respond appropriately to the comments of Reviewer 1, in particular,the "Generalizability" section. Please expand this section, giving a more detailed explanation (i.e. providing examples or evidence of how study findings can - or can not - be applyed to a different setting).

Reviewers' comments:

Reviewer's Responses to Questions

**Comments to the Author**

1. If the authors have adequately addressed your comments raised in a previous round of review and you feel that this manuscript is now acceptable for publication, you may indicate that here to bypass the “Comments to the Author” section, enter your conflict of interest statement in the “Confidential to Editor” section, and submit your "Accept" recommendation.

Reviewer #1: All comments have been addressed

Reviewer #2: All comments have been addressed

2. Is the manuscript technically sound, and do the data support the conclusions?

Reviewer #1: Yes

Reviewer #2: Yes

3. Has the statistical analysis been performed appropriately and rigorously? 

Reviewer #1: Yes

Reviewer #2: Yes

4. Have the authors made all data underlying the findings in their manuscript fully available?

Reviewer #1: Yes

Reviewer #2: Yes

5. Is the manuscript presented in an intelligible fashion and written in standard English?

Reviewer #1: Yes

Reviewer #2: Yes

6. Review Comments to the Author

Reviewer #1: The authors addressed all comments and improved their manuscript in large part.

However the "Generalizability" section is shallow and gives no actual information. Please expand this section, giving a more detailed explanation (i.e. providing examples or evidence of how study findings can - or can not - be applyed to a different setting)

Reviewer #2: good job, the manuscipt deserves publication. I have no more comments or suggestions or concerns to raise

7. PLOS authors have the option to publish the peer review history of their article (what does this mean?). If published, this will include your full peer review and any attached files.

Reviewer #1: **Yes: **Antonio Vinci

Reviewer #2: No

---

## [Author Response · Author response to Decision Letter 1]

22 Dec 2021

Responses to editor and reviewers 

Dear academic editor and reviewers,

Thank you for your comments and suggestions, which have crucial contribution to the manuscript. We have incorporated the comments and the details are given here.

Response to editor

1. The manuscript is in line with PLOS ONE requirements

2. Reference list is complete and correct

Response to reviewer 1

1. We were requested to provide detailed explanations on the generalizability of the study findings and we did it accordingly.

---

## [Decision Letter · Decision Letter 2]

17 Jan 2022

The status of prehospital care delivery for COVID-19 patients in Addis Ababa, Ethiopia: The study emphasizing  adverse events occurring in prehospital transport and associated factors

PONE-D-21-17807R2

Dear Dr. Ararso Baru,

We’re pleased to inform you that your manuscript has been judged scientifically suitable for publication and will be formally accepted for publication once it meets all outstanding technical requirements.

Kind regards,

Filomena Pietrantonio

Academic Editor

PLOS ONE

Additional Editor Comments (optional):

All comments have been addressed and the paper is now suitable for publication. The second reviewer had already communicated that the paper was suitable for publication

Reviewers' comments:

Reviewer's Responses to Questions

**Comments to the Author**

1. If the authors have adequately addressed your comments raised in a previous round of review and you feel that this manuscript is now acceptable for publication, you may indicate that here to bypass the “Comments to the Author” section, enter your conflict of interest statement in the “Confidential to Editor” section, and submit your "Accept" recommendation.

Reviewer #1: All comments have been addressed

2. Is the manuscript technically sound, and do the data support the conclusions?

Reviewer #1: Yes

3. Has the statistical analysis been performed appropriately and rigorously? 

Reviewer #1: Yes

4. Have the authors made all data underlying the findings in their manuscript fully available?

Reviewer #1: Yes

5. Is the manuscript presented in an intelligible fashion and written in standard English?

Reviewer #1: Yes

6. Review Comments to the Author

Reviewer #1: All comments have been successfully addressed by the Authors and I believe the paper is suited for pubblication.

7. PLOS authors have the option to publish the peer review history of their article (what does this mean?). If published, this will include your full peer review and any attached files.

Reviewer #1: **Yes: **Antonio Vinci

---

## [Editor Report · Acceptance letter]

24 Jan 2022

PONE-D-21-17807R2 

The status of prehospital care delivery for COVID-19 patients in Addis Ababa, Ethiopia: The study emphasizing  adverse events occurring in prehospital transport and associated factors 

Dear Dr. Baru:

I'm pleased to inform you that your manuscript has been deemed suitable for publication in PLOS ONE. Congratulations! Your manuscript is now with our production department. 

Kind regards, 

on behalf of

Dr. Filomena Pietrantonio 

Academic Editor

PLOS ONE